# An Overview of Mapping Quantitative Trait Loci in Peanut (*Arachis hypogaea* L.)

**DOI:** 10.3390/genes14061176

**Published:** 2023-05-28

**Authors:** Fentanesh C. Kassie, Joël R. Nguepjop, Hermine B. Ngalle, Dekoum V. M. Assaha, Mesfin K. Gessese, Wosene G. Abtew, Hodo-Abalo Tossim, Aissatou Sambou, Maguette Seye, Jean-François Rami, Daniel Fonceka, Joseph M. Bell

**Affiliations:** 1Department of Plant Biology and Physiology, Faculty of Sciences, University of Yaounde I, Yaounde P.O. Box 337, Cameroon; 2Department of Plant Science, College of Agriculture, Wolaita Sodo University, Sodo P.O. Box 138, Ethiopia; 3UMR AGAP, CIRAD, F-34398 Montpellier, France; 4AGAP Institute, Institut Agro, CIRAD, INRAE, University of Montpellier, F-34060 Montpellier, France; 5Centre d’Etudes Régional Pour l’Amélioration de l’Adaptation à la Sécheresse (CERAAS/ISRA), Route de Khombole, Thiès BP 3320, Senegal; 6Dispositif de Recherche et de Formation en Partenariat, Innovation et Amélioration Variétale en Afrique de l’Ouest (IAVAO), CERAAS, Route de Khombole, Thiès BP 3320, Senegal; 7Department of Agriculture, Higher Technical Teachers Training College, University of Buea, Kumba P.O. Box 249, Cameroon; 8Department of Horticulture and Plant Science, College of Agriculture and Veterinary Medicine, Jimma University, Jimma P.O. Box 378, Ethiopia

**Keywords:** QTL mapping, quality traits, interspecific, genetic, breeding, peanut

## Abstract

Quantitative Trait Loci (QTL) mapping has been thoroughly used in peanut genetics and breeding in spite of the narrow genetic diversity and the segmental tetraploid nature of the cultivated species. QTL mapping is helpful for identifying the genomic regions that contribute to traits, for estimating the extent of variation and the genetic action (i.e., additive, dominant, or epistatic) underlying this variation, and for pinpointing genetic correlations between traits. The aim of this paper is to review the recently published studies on QTL mapping with a particular emphasis on mapping populations used as well as traits related to kernel quality. We found that several populations have been used for QTL mapping including interspecific populations developed from crosses between synthetic tetraploids and elite varieties. Those populations allowed the broadening of the genetic base of cultivated peanut and helped with the mapping of QTL and identifying beneficial wild alleles for economically important traits. Furthermore, only a few studies reported QTL related to kernel quality. The main quality traits for which QTL have been mapped include oil and protein content as well as fatty acid compositions. QTL for other agronomic traits have also been reported. Among the 1261 QTL reported in this review, and extracted from the most relevant studies on QTL mapping in peanut, 413 (~33%) were related to kernel quality showing the importance of quality in peanut genetics and breeding. Exploiting the QTL information could accelerate breeding to develop highly nutritious superior cultivars in the face of climate change.

## 1. Introduction

Peanut (*Arachis hypogaea* L.) is a grain legume mainly grown in the tropics, subtropics, and warm temperate regions of the world. The genus *Arachis* originated in South America and all of its species produce their fruit underground [1]. Peanut is a self-pollinated, segmental allotetraploid [2,3], with 2n = 4x = 40 chromosomes [3,4]. It is an oilseed crop with global importance to food and nutritional security and a source of livelihood for millions of smallholder growers of Asia and Sub-Saharan Africa. World production of peanut was approximately 54 million metric tons (MT) harvested from 30 million hectares (Mha) in 2020. China is the world’s largest producer with 18 million metric tons (MT). Africa accounts for 32 % of worldwide production and the annual production and harvested area were 17 MT and 17.43 Mha, respectively, in 2020 [5]. Peanut ranked fifth among vegetable oilseed crops in terms of edible oil production (6.26 MT), preceded respectively by sunflower seed (21.56 MT), rapeseed (27.85 MT), soybean (57.74 MT), and palm (72.77 MT) [6].

Peanut is a major oilseed crop used for a variety of purposes, such as direct consumption of the seed (kernel), which can be eaten raw, roasted, boiled, or processed into confectionary and peanut flour for flavor enhancement, or crushed to edible oil. Nutritionally, it is a source of high-quality edible oil (35–60%), protein (22–30%), carbohydrates (10–25%), vitamins (E, K, and B complex), and minerals (Ca, P, Mg, Zn, and Fe) [7]. Peanut oil contains about 12 different kinds of fatty acids (FAs), with oleic acid (C18:1) and linoleic acid (C18:2) accounting for nearly 80% of the total [8]. The presence of relative proportions of various FAs affects the nutritional quality, flavor, and shelf-life of peanut seeds and products. The high linoleic acid content in peanut oil induces low oxidative and frying stability, resulting in rancidity, off-flavors, and short shelf life in produced foods [8,9]. Compared to a normal ratio, a high oleic-to-linoleic ratio leads to longer shelf-life and improved flavors [10]. The main consumption and production constraints of the crop include drought, pests, diseases, and environmental changes. These constraints have an impact on the content of oil and protein present, as well as the oil’s composition, which also has an indirect impact on the oil’s shelf life, aroma, flavor, cooking quality, and cooking time [4].

According to genetic, cytogenetic, phylogeography, and molecular data, the cultivated peanut, *A. hypogaea* is an allotetraploid, derived from hybridization between the diploids, *A. duranensis* (*A genome*) and *A. ipaensis* (*B genome*) [3,11]. These two species are members of the section *Arachis* [12]. Molecular analysis has shown that cultivated peanut has a limited polymorphism at the DNA level due to the crop origin, from single to a few hybridization events and the transition from diploid to tetraploid [1]. Various studies have revealed that wild relatives harbor high levels of genetic variation [13,14] that can be used for improving cultivated peanut. Several accessions of those species have been used to board the genetic diversity of cultivated peanut in different studies for a variety of traits including abiotic stress [15], disease resistance [16,17], oil content, and composition [18].

Chromatin introgression of a tiny fraction of the wild species genome while maintaining the genome background of the cultivated peanut is a means to discover the most untapped wild genes/alleles. As several authors have mentioned, direct gene transfer from wild diploid species has been hampered by ploidy differences, fertility barriers caused by species incompatibilities, linkage drag of desirable wild alleles with those conferring agronomically unadopted traits and, finally, difficulties in confirming hybrid identities and tracking introgressed segments [16,19,20]. These issues have been partly resolved by the production of synthetic allotetraploid that can be crossed with cultivated peanut [21,22,23]. as well as a number of molecular markers and genomics tools to ease introgression and genetic analysis [24]. GPBD 4, Span cross, Tamnut 74, TxAG 7, COAN, NemaTAM, and Tifguard are improved germplasm/cultivars that were developed using genes from wild *Arachis* species [13,24]. Among them, GPBD4 resulting from a cross between *A. hypogaea* and *A. cardenasii* derived introgression line has been widely used for improving disease resistance in a variety of breeding programs [25].

Mapping quantitative trait loci (QTL) and identifying markers that are linked to target traits are important steps toward accelerating the rate of genetic gains in breeding programs. QTL mapping is a routine technique used to identify genetic loci governing traits of interest [26]. QTL mapping in family-based populations requires (i) the development of appropriate mapping population and traits phenotyping; (ii) selection of appropriate molecular marker(s) and generation of molecular data with an adequate number of uniformly-spaced polymorphic markers; (iii) construction of genetic linkage maps to locate QTL using statistical programs.

The success of QTL mapping depends on the size of the mapping populations and the quality of the genotyping and phenotyping data. The availability of a tremendous number of genomic resources, including molecular markers, and genetic and physical maps have greatly eased the mapping of QTL and/or genes [27,28]. Many QTL have been reported for seed and pod-related traits [15,29,30,31,32,33,34,35], fresh seed or seed dormancy [36,37,38,39,40], nutritional quality traits [18,41,42,43,44,45,46,47], drought resistance [48,49,50], and disease resistance [51,52,53,54,55,56], providing potential tools for peanut improvement. However, few QTL are being effectively utilized in peanut improvement programs to produce elite cultivars.

The general aim of this paper is to review the recently mapped QTL in peanut, with a particular emphasis on traits related to kernel quality. Genome-wide distribution of QTL and their effective use in innovative schemes of marker-assisted selection (MAS) in peanut breeding are discussed.

## 2. Mapping Populations of QTL in Peanut

Genetic mapping populations are required for marker–trait associations both for oligogenic and polygenic traits. Genetic mapping can be broadly divided into two types: (i) family-based mapping, which is conducted on offspring of biparental or multiparent crosses, and (ii) natural population-based mapping, also known as association or linkage disequilibrium mapping, which is conducted on unrelated natural populations [26].

### 2.1. Family-Based Mapping Populations

The biparental mapping population is a kind of family-based mapping population created by crossing two parents that differ on the target of interest [57]. F2 [29,36,43,55,58], backcrosses (BC) [15,18,59,60,61,62,63], recombinant inbred lines (RIL) [44,45,46,47,48,49,50], chromosome segment substitution lines (CSSL) [64,65,66], and near-isogenic lines (NIL) [67] are the examples of biparental mapping populations. In peanut, F2, RIL, advanced backcross (AB) populations, and CSSL have been used for QTL mapping (Figure 1). The relative simplicity of construction, the high QTL detectability, and the low rate of linkage disequilibrium decay within chromosomes are the three main benefits of biparental populations. However, there are two main limitations of biparental populations: a lack of mapping precision due to the limited amount of effective recombination that occurs during population development and low genetic diversity because of the genetic bottleneck caused by the selection of two founders [68].

To overcome the limitations of biparental populations, multiparental mapping populations have been created; these populations increase the recombination rates and the genetic pool, leading to higher-resolution genetic maps [68,69,70]. Usually, they are suitable for high-resolution QTL mapping, although they also have some limitations, such as labor-intensive crossing, managing large population sizes, being time-consuming, and requiring significant investments in phenotyping and genotyping (Table 1). Nested association mapping (NAM) and multi-parent advanced generation inter-crossing (MAGIC) [71], are the two most important multiparent mapping populations [68]. Four NAM populations, ICGV 91114 and 22 genotypes, and ICGS 76 and 21 genotypes, using Florida-07 and Tifrunner as common parents with eight genotypes, were developed at ICRISAT and by USDA ARS and the University of Georgia (UGA), respectively [69,70,72]. Using the last two populations, a total of 42 SNP markers linked with 100-pod weight and 100-seed weight were detected [73]. Three MAGIC populations also were developed by crossing eight parental genotypes targeting multiple traits, including fresh seed dormancy, oil content, aflatoxin, and drought resistance [69]. Despite the discovery of QTL mainly using linkage analysis, natural population-based mapping has also been used in a couple of studies [74,75,76,77,78,79,80]. It is discussed in the next section.

**Figure 1 genes-14-01176-f001:**
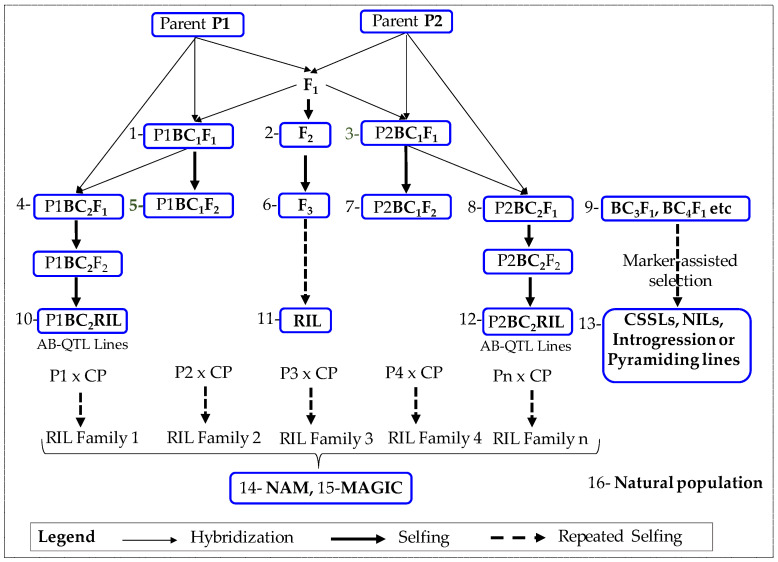
Derivation of biparental and other populations used for peanut. F2 [29,36,43,55,58], BC2F2 [15], BC1F1 [59], BC2F4 [60,61,62], BC2RIL [18], BC2F1 [63], RIL [44,45,46,47,48], CSSL [64,65,66], NIL [67] MAGIC [69,71], NAM [69,70,73], Natural population [74,75,76,77,78].

### 2.2. Natural-Based Mapping Populations

Natural population-based mapping is a method of mapping QTL that takes advantage of historic linkage disequilibrium to link phenotype to genotype by sampling distantly related individuals. However, the method comes with limitations: it is predominantly influenced by unknown population structure, leading to spurious associations, and also requires very large samples to have sufficient power to detect genomic regions of interest [74,75,76]. Natural population-based mapping utilizes diverse germplasm sets with high variability for economically important traits in a crop species with the advantages, in such cases, of high resolution and high allelic richness, with no investment in crossing. The main steps in natural population-based mapping include (i) collection of a sample population including elite cultivars, landraces, wild relatives, and exotic accessions; (ii) phenotyping target traits, estimation of broad-sense heritability, genotyping the population; (iii) quantification of LD extent of the selected population; (iv) identification of the influence of population structure and kinship; and (v) testing the association between genotype and phenotype using appropriate statistical approaches and validation of detected QTL [26,28].

Using this approach, several QTL have been reported in peanut. Two functional single nucleotide polymorphism (SNP) markers for two fatty acid desaturases (FAD2 for oleic acid, linoleic acid, and oleic-to-linoleic ratio) were found by phenotyping for quality traits and genotyping of the US “Mini Core Collection” using 81 SSR markers [74]. A total of 50 SSR markers linked with oil content, protein content, oleic to linoleic acid ratio, and fatty acid concentrations, with phenotypic variation explained (PVE) from 5.81 to 47.45 percent, were detected using 300 genotypes. Additionally, 12 QTL that linked to seed length and seed width, explaining phenotypic variance from 11.81 to 30.09 percent were also detected using these genotypes [75]. The other 107 significant SNP markers underlying pod weight, pod length, pod width, seed length, seed width, and 100-pod weight and 100-seed weight [76,79] were discovered using 158 and 250 genotypes, respectively. Similarly, 12 QTL associated with oil content were identified using 292 accession numbers [78], and 253 loci controlling oil content, protein content, oleic to linoleic acid ratio, and fatty acid composition were identified [77,80] using 120 and 250 genotypes, respectively. Among all the genotypes that have been used as a mapping population, more than 50% are from core collections. QTL that have been mapped for important quantitative traits, particularly utilizing a biparental approach, have been discussed in the next sections.

## 3. Molecular Markers for Linkage and QTL Mapping

A major application of molecular markers is the construction of linkage maps required for QTL mapping and marker-assisted breeding. Past molecular markers used in peanut include RFLP, RAPD, and AFLP. The first RFLP-based map for *Arachis* was created with 117 loci [81]. The first tetraploid RFLP-based genetic map was developed with 370 loci using an advanced population [60]. RAPD and RFLP markers were used to create a genetic linkage map using an interspecific diploid BC population derived from *A. stenosperma* x (*A. stenosperma* x *A. cardenasii)* [59]. In this study, 167 RAPD and 39 RFLP loci were mapped to 11 linkage groups, covering 800 cM. AFLP markers were also used to find DNA markers linked to aphid resistance and to construct a partial genetic linkage map of cultivated peanut [82]. Currently, the most commonly used molecular markers in peanut include SSR, SNP, and DArT. In 2009, the first genetic map based on SSR markers was constructed with 135 loci [83]. In the same year, 298 loci were mapped in 21 linkage groups, spanning a total map distance of 1843.7 cM in an advanced backcross population [84]. Among many SSR-based maps, these were developed for populations derived from TG26 and GPBD4, Sun Oleic 97R, NC94022, Tifrunner and GT-C20, and amphidiploid ‘’TxAG-6‘’ and ‘’Florunner”, to locate the genomic region underlying seed quality traits [18,41,42]. The maps contain 45 to 378 SSR loci spanning 671.1 to 2487.4 cM distance. In addition, with these maps, approximately 33 SSR-based genetic maps have been developed to date to identify the genomic areas responsible for disease resistance, drought resistance, yield, and yield component traits as reviewed previously [7,27,72].

SNP are the most common molecular markers in the genome, and they can be analyzed with high-throughput genotyping techniques. The first SNP-based genetic map in peanut was created with 1621 SNPs and 64 SSR markers on 20 linkage groups [85]. Aiming at oil and protein content and oil composition, three linkage maps were constructed with 2266–4561 SNP loci spanning a total map length ranging from 2032.39–2586.37 cM derived from Huayu28 and P76, Xuhua 13 and Zhonghua 6, and Yuhua 15 and W1202 [44,45,47], on 20 linkage groups. To date, SNP-based maps have been presented in peanut [7,27,72]. By using DArT markers, five genetic maps were constructed using F2 and advanced backcross populations. A genetic map using the F2 population derived from ICGV 00350 and ICGV 97045 has 1152 loci spanning a map distance of 2423.12 cM and a map density of 2.96 cM/loci developed [36]. A total of 854 (ICGV 07368 and ICGV 06420) and 1435 (ICGV 06420 and SunOleic 95R) marker loci were used to create two genetic maps, with total map distances of 3526 and 1869 cM, respectively [43]. The other two additional genetic maps, with 253 DArT and five SSR loci, and 1035 DArT and eight SSR loci, covering 1415.7 and 1500.8 cM of map length, respectively, were created using advanced backcross populations [63].

The availability of genome sequencing for peanut speeds up the development of different types of genotyping platforms/assays, including Kompetitive Alelle Specific PCR (KASP) assays, Golden Gate assays, Vera-code assays, micro-array-based markers, next-generation sequencing (NGS)-based markers, genotyping by sequencing (GBS), InDel markers and Affymetrix axiom SNP array. All these genotyping platforms are SNP-based since SNP markers are considered markers of choice and are amenable to high-throughput genotyping for several applications including QTL mapping.

## 4. Mapping of QTL

A QTL is a genomic region that is responsible for the quantitative variation of a trait. A quantitative trait is a measurable attribute based on the combined activity of one or many genes and their interactions with the environment, which can vary between individuals over a given range to generate a continuous distribution of phenotypes [86]. QTL mapping is important for identifying responsible genes, understanding variation mechanisms, determining how many QTL contribute significantly to the trait, determining how much variation is due to additive, dominant and epistatic effects, and determining the nature of the genetic correlation between different traits in a genomic region [87]. The steps involved in biparental QTL mapping are presented (Figure 2). To date, quantitative or metric traits in peanut include traits related to yield and yield component traits, flowering, agro-morphology, seed dormancy, quality and nutritional traits, and resistance to viral, bacterial, and fungal diseases.

### 4.1. Mapping QTL for Seed Quality Traits

Important quality traits that can be assessed by biochemical analysis of the peanut kernel include oil, protein, and sugar content, as well as fatty acid (FA), amino acid, and carbohydrate composition. The proportion of different FAs, such as saturated, monounsaturated, and polyunsaturated (PUFA), present in the oil determines the nutritional quality, flavor, and shelf life of both peanut kernels and products [8]. In particular, the concentration of oleic acid is one of the most important quality traits because it can increase the shelf life of peanut products and is beneficial for human health [88].

QTL mapping for traits related to oil and protein content as well as fatty acid composition in peanut has been reported [41,42,43,45,46,47,89]. Furthermore, QTL for unsaturated FA and the oleic acid to linoleic acid ratio [44], as well as QTL for saturated fatty acid composition [90], were also reported. A mapping population of 146 recombinant inbred lines (RILs) generated from a cross of TG26 x GPBD4 was used to discover QTL for protein, oil, oleic, and linoleic acid content, and for the oleic acid to linoleic acid ratio [41]. As the same authors have mentioned, GPBD4 has a desirable combination of early maturity, high yield, high pod growth rate, desirable pod and kernel features, high oil, and protein content, and an optimum oleic/linoleic acid (O/L) ratio, whereas TG26 is a semi-dwarf, erect cultivar with high linoleic acid content. Although the genetic map has low coverage (45 SSR markers on eight linkage groups), the authors reported 17 QTL on four genomic regions, including two major QTL for protein content. Likewise, several QTL were identified using two genetic maps developed from RIL populations derived from the crosses between Sun Oleic 97R and NC94022 and between Tifrunner and GT-C20 [42]. They found two major QTL for oil content on chromosomes A05 and A08 and 11 major QTL for oleic acid, linoleic acid, and the ratio of oleic acid to linoleic acid on the homeologous chromosomes A09 and B09. Using these mapping populations, 16 major QTL on B04 and A09/B09 were identified for palmitic acid, stearic acid, arachidic acid, gadoleic acid, behenic acid, and lignoceric acid content [90]. One consistent QTL for oil content was mapped on chromosome B03, explaining 14.36% of phenotypic variance [89]. Likewise, two genetic maps were developed using two F2 mapping populations, one for fatty acid composition (FA-population, ICGV 06420 x Sun Oleic 95R) and the other for oil content (OC-population, ICGV 07368 x ICGV 06420) [43], with 1435 and 854 SNP loci spanning 1869 and 3526 cM distances, respectively. In these two maps, 23 major QTL were identified on 11 genomic regions; 2 for oil content and 21 for fatty acid composition variation, explaining up to 41% of PEV.

Another high-resolution genetic map, with 2334 SNP markers and a total length of 2586.37 cM, was constructed using a RIL population developed from the cross between high- and normal oleic cultivars [44]. The authors reported 29 major QTL for oleic and linoleic acid content as well as oleic to linoleic acid ratio, explaining 10 to 57.6% of the phenotypic variance, which were mapped on chromosomes A03 and A09/B09.

More recently, 14 major QTL involving oil content on A05, A06, A08, B06, and B10, explaining up to 27.19% PEV, were discovered from the three mapping populations derived from Xuhual13 and Zhonghua6, Yuhual15 and W1205, and Zhonghua10 and ICG12625 [45,46,47]. Moreover, major QTL associated with protein stearic acid, behenic acid, and arachidic acid contents were mapped on chromosomes A05, A06, and A08 [47]. The locations of the chromosomes where the aforementioned QTL are located are shown in Figure 3.

We performed a comparative QTL analysis using data from the studies above in order to gain more insight into the genome-wide distribution of kernel-quality QTL and to document the most consistent ones for future use in marker-assisted breeding. The map location of the QTL is presented in Figure 3 and the detailed data of all the 1261 QTL reported in this review, including information on the 413 quality-related QTL are found in Appendix A. Except for chromosome B01, QTL related to quality traits are mapped on 19 out of the 20 peanut chromosomes. We found that QTL for quality traits are mainly clustered on chromosomes A05, A08, and A09 for the A genome, and B04, B08, and B09 for the B genome. For instance, many QTL for oil and protein content as well as fatty acid compositions (arachidic, arachidonic, behenic, stearic, palmitic, linoleic, and oleic) co-localized on chromosome A05 and were consistent among environments (Figure 3 and Appendix A). Furthermore, QTL for oleic acid, linoleic acid, and the oleic/linoleic ratio from different studies were found in common genomic regions on chromosomes A05, A08, A09, B04, and B09. In chromosome B09, the common QTL are closely linked to markers ahFAD2B and SNP markers, Marker2575339 or Marker239598. The AhFAD2B QTL, on chromosome B09, explained up to 57% of phenotypic variation of oleic acid or linoleic acid content [42,44]. Similarly, the AhFAD2A and Marker4391589 or Marker4463600 on chromosome A09, are common among studies and explained up to 29% of phenotypic variation [42,44]. Additionally, AhMXZ190701 was discovered to be tightly linked to a major and stable QTL A08 for oil content [42,45]. These consistent markers, AhMXZ190701, ahFAD2B, ahFAD2A, Marker2575339, or Marker2379598 have been used for QTL validation and MAS of quality traits [45,91]. Likewise, several QTL for arachidic, behenic, stearic, palmitic, linoleic, and oleic acid and oil content, mapped in three studies are linked to the marker RN34A10 on chromosome A7 (Figure 3) [42,90]. Furthermore, consistent QTL among traits and environments were also reported. From a QTL mapping study on four environments, among the 110 QTL related to nine quality traits, 36 pleiotropic QTL were associated with two or more traits and showed consistent effects in more than one environment [47].

The consistent QTL identified for peanut quality traits, thanks to published studies, reviewed here can be used in breeding special-purpose peanut cultivars. However, some QTL need to be validated with fine mapping considering their positions on chromosomes differed in different studies, probably due to the genetic material, large QTL intervals, and statistical imprecisions.

### 4.2. Mapping QTL for Agro-Morphological Traits

In order to meet the food needs of a growing world population, the main goal of the plant breeding program has been to increase pod yield. In this paper, SSR, SNP, and DArT markers linked to agro-morphological traits utilizing various mapping populations, including F2, BC2F1, BC2F3, BC3F2, BC4F3, and recombinant inbred lines (RILs), have been discussed. Given this, a total of 266 main-effect QTL were mapped for pod- and seed-related traits: 100-seed weight, 100-pod weight, pod weight, pod length, pod width, seed length, seed width, and pod number, using F2 [36,89] and RIL [31,32,33,34,35,92,93] populations. A total of 44 QTL for 100-pod weight were identified, explaining up to 38.15% of the variance on chromosomes A05, A07, A08, B02, B03, B07, and B08, as well as homeologous chromosomes A07 and B07 [30,31,89]. On chromosomes A02, A03, A04, A05, A06, A07, A08, B02, B03, B04, B05, B06, and B08 with A05 and B05 homoeologous loci, 35 QTL were reported with 5.68 to 35. 9% phenotypic variance explained linked with 100-seed weight [31,32,89]. These 05A1430-A05A1601, A05A1344-A05A1562, and A05A1430-A05A1601 major and stable SSR markers increased pod length, pod width, and 100-pod weight by 27.84, 14.12, and 26.82%, respectively [30]. Similarly, major QTL for pod weight and seed weight were reported [35].

Along with traits related to seeds and pods, QTL for flowering, plant height, and fresh seed dormancy were also identified. A total of 30 QTL were reported, ranging in PVE from 1.15 to 21.82%, which underlie the days for 50% flowering and the first days of flowering, using three genetic maps created from TAG 24 x GPBD 4, and TAG 24 x ICGV 86031 [32,94]. For plant height, 71 main-effect QTL [32,92,95], were identified, accounting for up to 26.27% of the phenotypic variance. Several main-effect QTL linked to seed dormancy or fresh seed dormancy in peanut have recently been reported to explain up to 71.21% of the phenotypic variance using F2 and RIL populations [36,37,38,39,40]. For fresh seed germination, QTL were detected on seven homoeologous chromosomes, which are in both A and B genomes with one major stable marker [38]. The co-localization of QTL for the studied traits was reported. Several QTL associated with plant architecture as growth habit and plant height co-localized with those associated with flowering [96]. Moreover, the QTL of yield components, such as 100-pod weight, pod weight, and pod length [30,33], 100-pod weight, 100-seed weight, and pod weight [29,32], were co-localized on chromosomes A05 and A07. Overall, the QTL mapped related to agro-morphological traits include QTL related to plant architecture, flowering, fresh seed or seed dormancy, and yield component traits. 

In this review, QTL underlying drought resistance traits were also highlighted. For these drought resistance traits, shoot dry weight, transpiration efficiency, leaf area, transpiration rate, transpiration, and SPAD chlorophyll meter readings (SCMR), 127 QTL were discovered with the phenotypic explained variation ranging from 4.2 to 22.09% [48,49,50]. From a field experiment using well-watered and water-limited treatments a total of 13 QTL, individually explained 10.4%–20.1% of the phenotypic variance, were significant for the stress tolerance indices (STI): two for total biomass on chromosomes B06 and A05, one for pod weight on chromosome A05, one for seed weight on chromosome A05, two for haulm weight on chromosomes A02 and A05, two for 100 pod weight on chromosomes B02 and B05, and two for 100 seed weight on chromosome A05. In most cases, the STI-related QTL co-localized with the yield component-related trait for which they were calculated [15]. Main-effect QTL reviewed for the important traits are found in Table 2.

### 4.3. Mapping QTL for Disease Resistance Traits

The most efficient and environmentally friendly way to fight against pests and diseases to control yield losses is to develop disease-resistant cultivars by finding the responsible QTL or genes. For these particular important quantitative traits, 134 QTL were reviewed using RFLP, SSR, SNP, and DArT markers with F2, BC2F1, BC3F1, BC2F4, and RILs, mapping populations. Of the 134 QTL, 82 have been found to be linked to resistance to late leaf spot, early leaf spot, rust, smut, and bacterial wilt using F2 [55,58], and RILs [51,52,53,54,56,97], mapping populations with SSR and SNP markers. Using 103 RIL genotypes derived from a cross between JS17304-7-B and JS1806, 10 QTL underlying smut resistance were identified with a phenotypic variance of up to 11.4% [56]. A total of 12 main-effect QTL linked to rust resistance traits were identified in a RIL population [52,55]. From a RIL mapping population derived from Tamrun 0L07 and Tx964117, eight QTL were identified as linked to leaf spots that explained phenotypic variance ranging from 8 to 20% [97]. Similarly, 36 QTL that linked early and late leaf spots were discovered using a 192 RIL population produced from a hybrid of Florida-7 x GP-NCWS16 cultivars [53,54]. Three minor effects of QTL explaining up to 0.26% PEV for bacterial wilt were discovered using the mapping population derived from the cross of Yueyou 92 and Xinhuixiaoli [58]. Given this review, QTL linked to disease resistance traits have been mapped on all linkage groups except A04 and B07 having resistance QTL from the donor parents on A02/B02, A03, B03, and B05 for late leaf spot, rust, and smut [53,54,55,56]. Homoeologous QTL were discovered on A05/B05 for late leaf spot [53,54], and on A02/B02 for rust [55]. QTL on A02, B03, and A05 are common among studies and common for both disease and yield component traits, which is also supported by their strong genetic correlation.

### 4.4. QTL Mapping Using Interspecific Synthetic Tetraploids

From the reported relevant studies, we found that several interspecific populations have been used for QTL mapping in peanut. Those populations are developed from crosses between synthetic tetraploids and elite varieties and allowed the broadening of the genetic base of cultivated peanut and helped with the mapping of QTL and identifying wild beneficial alleles for economically important traits. Indeed, cultivated peanut has low genetic variation due to its origin in a single hybridization event between two diploid species, followed by chromosomal doubling and crossing barriers with wild diploid species [60,98]. The low genetic variability for traits of importance and polyploidy is a bottleneck to peanut improvement. The primary gene pool of peanut includes mainly tetraploids such as cultivars, advanced breeding lines, and landraces of *A. hypogaea*, as well as *A. monticola* [22,87]. This gene pool is cross-compatible, allowing fertile hybrids to be produced. The secondary gene pool, on the other hand, consists of wild diploid species (2n = 2x = 20) [24,27], which possess desirable alleles for several economically important traits, such as biotic and abiotic resistance. Despite rich diversity with desirable alleles, the use of wild relatives has been limited in breeding programs because of ploidy differences with cultivated peanut [1,27]. However, the ploidy level difference-induced bottleneck has been solved using interspecific synthetic allotetraploids [22,23,60]. The effect of polyploidization and hybridization on various traits in *Arachis* interspecific synthetic tetraploids has been studied [20,62]. Wild chromatin introgression into cultigen from the synthetic tetraploid increase DNA polymorphism helping to map QTL by using the AB-QTL or CSSL analysis [15,17,18,63,98].

To the best of our knowledge, six synthetic tetraploids have been used to date for QTL mapping of traits related to disease resistance, drought resistance, agronomic traits, and oil quality traits utilizing BC2F1, BC3F1, BC2F3, BC2F4, BC3F2, BC4F3, and BC3F6 mapping populations (Figure 1). A total of seven QTL were identified from BC3F1 in the population developed from cultivated “Florunner” and synthetic TxAG-6 for root-knot nematode resistance using RFLP markers [17]. Using the BC3F6 generations of the mapping population developed from the aforementioned parents genotyped using SSR markers, 29 QTL associated with oil content, six fatty acid traits, and the oleic to linoleic acid ratio were detected on 20 genomic regions [18]. Of the 20, two are major and stable and linked to oil content and the oleic to linoleic acid ratio with phenotypic variance explained at 17–21 and 13–31% PEV, respectively. Another genetic mapping based on SSR markers was performed using a cross between the cultivated variety “Fleur 11” and the synthetic tetraploid AiAd derived from *A. ipaensis* x *A. duranensis,* the two diploid ancestors of the cultivated [84]. The advanced backcross populations (BC2F3 and B3F2) from this cross were utilized for QTL mapping of flowering date, pod weight, pod number, seed number, pod size, seed size, pod maturity, and biomass under well-watered and water-limited treatments, yielding a total of 95 QTL [15]. About half of the QTL positive effects were associated with alleles of the wild parent, highlighting that peanut wild relatives represent a reservoir of useful alleles for peanut breeding. In addition, by using BC4F3 (CSSL) lines from the same cross, the authors also found 42 QTL mapped for plant growth habits, the height of the main stem, plant spread, and flower color [64]. Likewise, using the same recurrent cultivated parent, Fleur 11 with a different synthetic parent ISATR52B, 38 QTL were identified underlying the flowering date, plant architecture, yield-related, pod, and seed morphology traits on 16 chromosomes [61]. They found that almost 50% of the positive QTL effects were associated with alleles of ISATR52B.

Furthermore, 28 QTL explaining 6.7–50.9% PEV linked to 100-seed weight, oleic to linoleic acid ratio, and late leaf spot and rust resistance were identified from the two interspecific populations derived from ICGV 8764 x ISATGR 265-5A and ICGV 91114 and ISATGR 1212 genotyped with SSR and DArT markers spanning 1415.7–1500.8 cM map length [63]. Seven QTL for late leaf spot and rust resistance and one for the oleic to linoleic acid ratio, with phenotypic variance explained up to 9.7 and 14.8 on chromosomes A01, A07, and A08, were found in the population derived from ICGV 91114 X ISATGR 1212. In the population derived from ICGV 8764 XISATGR 265-5A, three QTL were found for each trait for the oleic to linoleic acid ratio and 100-seed weight, up to 47% phenotypic variance, and 14 QTL were found for late leaf spot and rust resistance, up to 50.9% phenotypic variance. Of the 28 QTL, three contributed favorable alleles from wild genomic segments. In an effort to enhance foliar disease resistance, a cross between ICGS 76 and synthetic amphidiploid ISATGR 278-18 has been recently revealed [62]. In this population, 14 and 10 QTL associated with late leaf and rust resistance, up to 38.58% PEV, respectively, were identified.

Overall, synthetic tetraploids are used for alien chromatin introgression into cultivated peanut by resolving polyploidy differences and easing genetic and meiotic analyses and QTL detection for numerous economically important traits despite constraints, such as hybrid fertility and linkage drag. 

### 4.5. Clustering of QTL of Quality and Agronomic Traits

Of the 1261 main-effect QTL reviewed here, we identified relevant QTL related to quality traits that clustered with QTL of agronomic traits. For instance, the QTL of oil content associated with the SSR marker IPAHM103 on chromosome A3 with a PEV of 7.1–10.2% for oil content [41] is the same identified for rust resistance with a PEV of 6.9–55.2% [51] and late leaf spot resistance. Likewise, the SSR marker PM36 mapped for oil content in several studies [18,41] was identified in the QTL regions of pod and seed weight and shelling percentage in chromosome 5 in various studies [15,30,31,33].

As far as the oil quality traits are concerned, QTL (TC6H03–TC11A04, TC5A07–IPAHM395, and TC3A12–PM433) were common for both oleic and linoleic acid, which is also supported by their strong negative correlation [41]. Consistent QTL, ahFAD2A and ahFAD2B, IPAHM372-ahFAD2A, GM1840-ahFAD2B, and GNB377-ahFAD2A linked oleic acid, linoleic acid, and O/L ratio [42]. In addition to the SSR markers linked for oil quality traits, SNP markers Marker2575339 and Marker2379598 in B09 and Marker4391589 and Marker4463600 in A09 were associated with oleic acid, linoleic acid, and the ratio of oleic acid to linoleic acid (O/L) [44]. A major and stable QTL on A05, flanked by the markers bin1572 and bin1573 on 0–0.5 cM was detected and showed a negative additive effect on oil, palmitic, stearic, arachidic, and behenic acid content and positive additive effects on protein, oleic, arachidonic acid [47]. These stable oil-related QTL on A05 is quite common to a stable and major genomic region on A05 that has been reported for pod and seed-related traits in several studies [30,33]. The co-localized interval on A05 was located on 1.3 cM (99.50–99.78 Mb) by the flanking markers Ad05A20262 and AHGA160418 and harbored the major QTL for pod length, pod width, and 100-pod weight with 17.97–43.62% of phenotypic variations [33]. For these traits, from another QTL mapping, three more major QTL co-located in about 2.47 Mb genomic region of the A05 with 13.75 to 26.68% PVE by the flanking markers A05A1430-A05A1601 [30]. Moreover, three major QTL common for pod length and seed length on A05 with up to 26.11% PEV were identified [29]. The cluster of many major QTL detected on A05 in different studies for oil content and seed or pod-related traits suggests it may harbor important genes controlling these traits, which can be used, simultaneously in marker-assisted breeding. This clustering also suggests linked QTL of these distinct traits or QTL with pleiotropic effects. Thus, breeding for an agronomic trait may indirectly, positively, or negatively affect a quality-related trait. 

### 4.6. Statistical Methods and Limitations of QTL Mapping in Peanut

QTL are, by definition, merely significant statistical associations between genotypic values and phenotypic variability among the segregating progeny. Statistical methods for family-based mapping include (i) single-marker analysis (SMA) used to identify QTL according to the difference between the average phenotypes of different genotype groups without linkage map; (ii) interval mapping (IM) based on maximum-likelihood parameter estimation and regression, which efficiently estimates the effect and position of a QTL within two flanking markers; (iii) composite interval mapping (CIM), to overcome such limitations of the IM method; (iv) inclusive composite interval mapping (ICIM) and (v) multiple interval mapping (MIM), an extension of IM, that tends to be more powerful and precise than CIM in identifying QTL and allows the estimation of multiple QTL with epistasis. A large number of software implementing the above methods are used in peanut, including R/qtl, QTL Cartographer, and ICIM Mapping [15,28,30,33,34].

From most studies reviewed here, QTL mapping technique accuracy depends on several factors, including the statistical method’s capacity to locate and estimate the genetic effect of the QTL, the type and size of the mapping population, the genetic and heritability of the trait, the number and contribution of each QTL to the total variance, their interactions and their distribution over the genome. In addition, the ploidy coupled with mixed meiotic behavior is not yet considered in QTL detection and may affect the accuracy of QTL mapping in peanut. Along with these accuracy factors, QTL analysis has limitations like other techniques. Some of these limitations include the inability to detect all loci, the number of QTL detected, their precise position, and their effects are subject to statistical error. Major QTL are often missed and epistatic effects and QTL environmental interactions are found in some cases. QTL mapping is often time-consuming and labor-intensive, requires in-depth knowledge about the function and genomics of the trait of interest, and incurs high costs for genotyping and phenotyping. The large size of QTL and the low resolution of mapping greater than 10 cM in size are some of QTL mapping’s population specificities. In many cases, more experiments are needed to confirm the results of QTL mapping. However, by using consistent QTL that have been mapped, it is expected that the next-generation crop varieties could be developed with enhanced quality traits, better yield, and disease resistance.

## 5. Toward More Effective Use of Marker Assisted Selection (MAS) and QTL in Peanut

One of the current challenges in peanut is to use QTL of interest to accelerate genetic improvement. Considering the constraints of phenotypic selection—labor-intensive, costly, and time-consuming—MAS has emerged as a potential tool to achieve rapid results with the help of molecular markers and QTL of interest in plant breeding [99,100]. There are different molecular approaches used under the umbrella of MAS, such as marker-assisted backcrossing (MABC), gene pyramiding, MARS, and GS [101,102]. Some of the innovative applications of MAS including combined MAS, marker-directed phenotyping, inbred or pure-line enhancement, single large-scale MAS, breeding by design, and Mapping As You Go (MAYG) have been published previously [103,104]. Here, some schemes of MAS have been highlighted and may be useful for peanut breeding.

### 5.1. Marker-Assisted Backcross Selection (MABC)

MABC is a technique that can be used to incorporate one or more QTL from a donor parent (DP) to a recurrent parent (RP), which is a superior variety but lacking the target trait. Four to six generations of backcrossing are required to introduce the QTL into an elite cultivar and recover the recurrent parent [103,105]. Foreground selection, recombinant selection, and background selection are the three basic steps in marker-assisted backcrossing. In foreground selection, the desired plant is chosen using markers linked to the target QTL. Random markers across the entire genome can be used to screen the recurrent parent genome in the background selection context [103,104,105]. Recombinant selection is a kind of foreground selection that aims to remove the DP genome flanking the target QTL to avoid the linkage drag brought on by the close linkage of some undesirable traits with the target trait from the DP. MABC has been used in peanut breeding. Two BC3F1 lines, TMG-29 and TMG-46, have shown enhanced resistance over the highly susceptible TMV 2 peanut variety of late leaf spot (LLS) and rust-resistant genotypes using resistant donor “GPBD 4” [106]. According to [107], the backcrossing lines that were created by introducing the two mutant alleles, ahFAD2A and ahFAD2B, into the high oil content breeding line ICGV06100 showed a 97% increase in oleic acid content in comparison to the recurrent parent. A total of 22 BC3F4 and 30 BC2F4 introgression lines for rust and late leaf spot resistance, as well as 46 BC3F4 and 41 BC2F4 for high oleic acid, were created by crossing the donor rust resistance parent, GPBD 4, with three susceptible peanut cultivars (GJG 9, GG 20, and GJGHPS 1) in order to develop rust-resistance and late leaf-spot-resistance and a high oleic acid content genotypes [108]. Recently, the high-oleic-acid BC4F6 line “YH61” was created after four backcrossing of “huayu22” with the donor “KN176” with a high-oleic-acid content [109]. Furthermore, chromosome segment substitution lines [64,65], near-isogenic lines [67], and AB-populations [15,17,18,61,62,63] use, in such cases, MABC approaches and all facilitate genetic analysis, QTL introgression, and variety development in a simultaneous manner. MABC strategy can be more effectively used for introgression of major-effect QTL controlling different economically important traits to develop improved varieties.

### 5.2. Marker-Assisted Recurrent Selection (MARS)

In marker-assisted recurrent selection (MARS), plant genotypes are selected with the help of molecular markers that have been linked to the genes or QTL of interest. Once markers that are tightly linked to QTL of interest have been identified breeders use specific DNA marker alleles as a diagnostic tool to identify plants carrying the QTL [86], the chosen individuals that have QTL of interest are then subjected to controlled pollination to create lines that have the best possible complement of QTL from both parents [101,110]. This scheme is less used and may be useful for developing breeding material displaying QTL for targeted breeding traits. MARS is more suitable for introgression of minor-effect QTL controlling different important traits. 

### 5.3. Marker-Assisted QTL Pyramiding (MAQP)

Pyramiding is the simultaneous integration of several QTL from multiple parents into a single genotype to create superior lines and varieties [101,102,111]. Marker-assisted QTL pyramiding can speed up the process by lowering the number of generations that the researchers must evaluate to ensure that they have the desired QTL combination [112]. The QTL pyramiding method consists of two fundamental steps: the QTL fixation step, which aims to fix the target QTL into a homozygous state, and the pyramiding step, which aims to accumulate all the target QTL into a single genotype known as the root genotype [113]. This breeding technique was used in peanut to develop nematode resistance and high oleic gene-containing genotypes [114]. It is expected that the next-generation crop varieties could be developed with enhanced quality traits, disease resistance, and better yield by using MAQP.

### 5.4. Genomic Selection (GS) by Using Know QTL

GS is a promising method for genetic improvement of complex traits that are regulated by many QTL, each of which has a small or main effect [72]. In addition to promising to address complex traits, the GS strategy offers the benefit of shortening the selection cycle and eliminating lengthy phenotyping by favoring superior lines based on the prediction of the genomic-estimated breeding values (GEBV). When phenotype data and information on markers known to be associated with known QTL were combined to calculate estimated breeding values (EBVs), the gains from selection in plant breeding experiments increase significantly [102]. In the same way, the targeted QTL were accumulated at a much higher frequency when known QTL were included in the GS model as compared to when the standard ridge regression was applied. Several factors, including the size of the training population and its constitution/structure, precision, and quality of phenotyping, marker density, and trait heritability, have an effect on the prediction accuracy of GS [102]. This approach has not yet been applied in peanut and could be helpful for peanut breeding by using both cultivated and wild relatives and known QTL in prediction models.

### 5.5. Combined MAS and Marker-Directed Phenotyping for Quality Traits

In comparison to MAS or phenotypic screening alone, MAS combined with marker-directed phenotyping increases genetic gain and may help identify undiscovered QTL [115]. This combined selection aids in the selection of traits when phenotyping is more expensive than genotyping. In most cases, there is a low level of recombination between QTL and marker, which means we cannot entirely rely on markers for selecting desirable phenotype traits. However, it will help reduce the number of plants to be evaluated, which reduces the cost of phenotyping. One of the successful examples to explain this scheme is the rice primary QTL sub 1, which controls submergence tolerance [116]. This scheme may be useful, mainly for quality traits in peanut such as fatty and amino-acid acid composition where phenotypic screening is costlier than marker genotyping.

## 6. Use of Mapped QTL in Peanut Genetic and Breeding

For the over 1261 QTL reviewed here, to the best of our knowledge, only less than 10 have been used in peanut breeding programs. The current challenge is to use validated mapped QTL for peanut breeding for the fast-track development of improved varieties. The common QTL found between different studies and different genetic materials, and those with high positive effects, as reported above, can be mobilized in the breeding program by making sure to identify the beneficial QTL allele and combining the ability of the parents. In addition, using mapped QTL for breeding requires fine mapping, QTL and markers validation, and marker-assisted selection. QTL validation often needs cross-verification of QTL in different populations or/and different environments and fine mapping. Marker verification needs testing of molecular markers in germplasm and identifying polymorphic markers. Polymorphic markers around the validated QTL could be used for an indirect selection to strengthen conventional breeding. We expect that the QTL, once validated, are deployed in molecular breeding programs aimed at enhancing targeted traits in peanut through MAS, genomic selection (GS), or holistic and innovative schemes.

## 7. Use of Lines with Beneficial QTL Alleles for Fast QTL Introgression and Variety Development

QTL analysis from the above studies identified QTL alleles with favorable effects on peanut breeding. Likewise, AB-QTL analysis identified several introgression lines with good agronomic, oil quality, and disease-resistance traits. In addition, AB-QTL and CSSL lines, in common cases, are designed to map and facilitate QTL introgression from unadapted germplasms such as landraces and wild species into elite lines [117]. There are several advantages such as the simplicity of mapping the population in phenotypes to the recurrent parent and reducing, in the process, deleterious alleles from the donor parent, the possibility of epistasis, and linkage drag. After QTL mapping, only one or a few generations are needed for identifying QTL-NILs [118]. Several AB-QTL populations [15,17,18,63], CSSL [64,65], and NIL [67] have been developed in peanut. Beneficial QTL alleles, carried by AB-QTL lines or CSSL have been identified and, could be utilized through introgression into the genetic backgrounds of cultivars used by producers. In some cases, AB-QTL, CSSL, or NIL outperformed cultivated varieties and meet market needs. Thus, they could be directly promoted as a new variety. For instance, recently in Senegal, six new varieties, Rafet car, Tosset, Komkom, Jambar, Yakaar, and Raw Gadu with high yield profiles were homologated from CSSL [64], 12CS_031, 12CS_069, 12CS_120, 12CS_068, 12CS_037, 12CS_028, respectively, derived from the cross between “Fleur11” and the wild synthetic tetraploid AiAd (*A. duranensis and A. ipaensis*). The next decade will see heavy use of these kinds of lines for the development of new varieties.

## 8. Conclusions and Perspectives

The chromosomal or genomic regions known as QTL are responsible for variation in a quantitative phenotype. Finding genomic regions with QTL, estimating the effect of the QTL on the quantitative trait, determining how much of the trait’s variation is due to a specific region, and discovering the gene action linked to the QTL are the objectives of QTL mapping. In this paper, we have reviewed 1261 QTL that govern economically important traits for peanut breeding that have recently been mapped through diverse sources of mapping populations including F2, recombinant inbred lines, advanced backcross populations, chromosome segment substitution lines, and nested association mapping (NAM) or multiparent advanced generation inter-crossing (MAGIC) populations, as well as a variety of molecular markers. Protein content, oil content, fatty acid composition, yield, yield component, drought resistance, and pest and disease resistance are considered significant and important quantitative traits. Various limitations of QTL mapping have been discussed, and the solutions proposed to overcome them are the constant development of molecular platforms, new genetic materials such as introgression lines that help in mapping the small effects, and sophisticated bioinformatics that can handle polyploidy issues and mixed meiotic behavior, false-positive results or statistical errors. Introgression of validated mapped QTL alleles, fruitfully associated with preferred traits, into the genetic background of the elite varieties is a current challenge for peanut breeding. Integration of high throughput phenotyping and new-generation phenomics tools with MAS could greatly accelerate progress in peanut genetic improvement. The present review discusses the current status and future scope of using mapped QTL for breeding purposes in peanut, which will cause not only an increase in the rate of developing climate-resilient superior cultivars but also help in providing vegetable oil and proteins to the growing human population worldwide.

## Figures and Tables

**Figure 2 genes-14-01176-f002:**
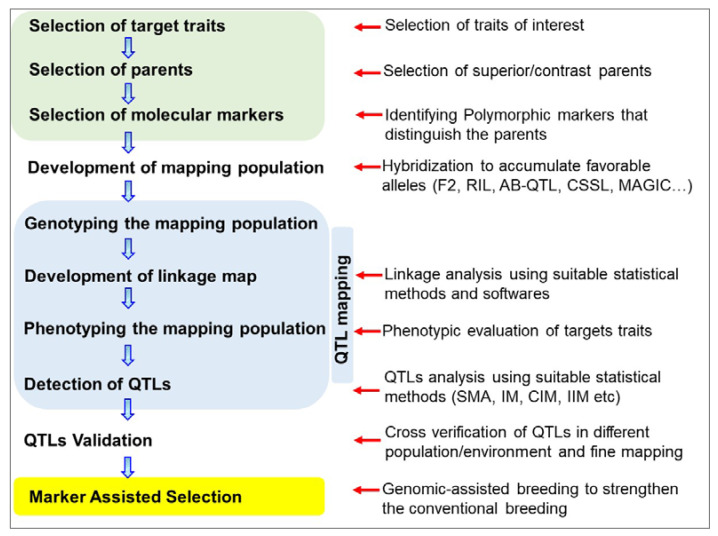
Steps involved in biparental QTL mapping and further use of the QTL in breeding programs.

**Figure 3 genes-14-01176-f003:**
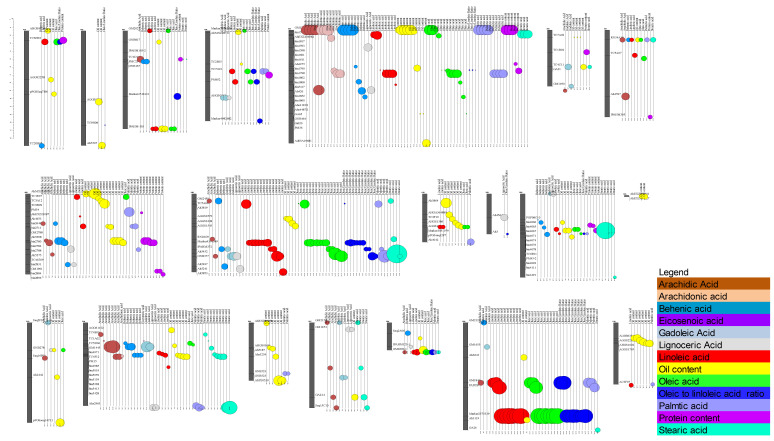
Schematic map of known QTL related to quality traits in peanut. The QTL detected are distributed over the GLs of the A genome, named from A1 to A10, and those of the B genome named from B1 to B10 indicated by gray-colored segments. Locus names related to the QTL are shown on the right of each GL. The QTL detected for each trait are indicated by colors in the legend. The peaks of the QTL are respectively indicated circles. The size of the circles is proportional to the phenotypic variance of the trait indicated by the authors.

**Table 1 genes-14-01176-t001:** Advantages and disadvantages of populations used in peanut mapping.

Populations	Advantages	Disadvantages	References
BC	- Useful for the introgression of wild chromatin and specific genes/QTL - Easy analysis	- Time requirement to construct in such cases - Estimating dominance effects is impossible - Not suitable for Meiotic behavior analysis	[15,59,60]
F2	- Require less time to construct - Impossible to determine the degree of dominance and additive effects - Easy analysis - Suitable for Meiotic behavior analysis	- Precision is low - Temporary nature - It is not repeated across years and locations	[2,43,55,58]
NIL	- Suitable for fine mapping - Useful for tagging the gene - Highly reliable and accurate statistically - Suitable for quantitative and qualitative trait tagging	- It takes time to construct - Not suitable for whole linkage mapping - Problem of linkage drag	[67]
RIL	- Abundance of recombination - Immortality: replicable throughout locations and years - Very useful in identifying tightly linked markers	- Impossible to estimate dominant effects - Time requirement: many seasons and generations are needed for the development	[44,45,46,47,48,49,50]
CSSL	- Immortality: replicable throughout locations and years - Very useful in identifying tightly linked markers	- Time requirement: many seasons and generations are needed for the development	[65]
NAM or MAGIC	- Several alleles than biparental populations - Several QTL segregating than biparental populations - Rapid fine mapping - Useful for candidate	- Time to construction/establish - Require more markers than biparental - Require larger population than biparental	[69,70,71,72,73]
Natural	- Available collections - High diversity - Natural recombination - When LD limited, Precise mapping	- Time to construction/establish - Require more markers than biparental and MAGIC or NAM population - Population structure - Spurious association - When high LD, coarse mapping - Rare alleles poorly identified	[74,75,76,77,78,79,80]

**Table 2 genes-14-01176-t002:** Main-effect QTL reviewed for the important traits of peanut.

Quality-Related Traits			
Traits Studied	QTL Identified	Phenotypic Variance Explained	References
Oil content	80	0.76–27.19	[18,41,42,43,46,47,89,90]
Protein content	22	0.76–26.99	[41,47]
Oleic acid	58	0.13–57.56	[18,41,42,43,44,47]
Linoleic acid	54	0.17–57.56	[18,41,42,43,44,47]
Oleic/linoleic acid ratio	32	1.04–43.41	[18,41,42,44]
Palmitic acid	32	0.3–34.35	[43,47,90]
Arachidic acid	32	0.13–36.93	[18,43,47,90]
Stearic acid	31	0;13–78.6	[18,47,90]
Behenic acid	32	0.76–26.99	[18,43,47,90]
Eicosanoid	1	0.2	[18]
Lignoceric acid	15	2.89–12.61	[43,90]
Gadoleic acid	16	2.55–15.11	[90]
Arachidonic acid	8	0.76–26.99	[47]
**Agro-Morphological-Related Traits**			
Traits Studied	QTL Identified	Phenotypic Variance Explained	References
Plant height	77	0.01–26.7	[15,32,92,95]
Hundred-pod weight	48	3.33–38.15	[15,30,31,33]
Fresh seed/seed dormancy	54	69.3–74.7	[36,37,38,39,40]
Days to flowering	31	1.15–21.82	[15,32,94]
Pod weight	20	7.7–29.7	[15,35,48,92]
Pod length	S2	1.25–26.46	[15,30,31,33,89]
Pod width	54	5.1–43.63	[15,29,30,31,33,89]
Seed length	32	3.03–20.8	[15,35,48]
Seed width	33	2.21–23.7	[15,29,31]
Harvest index	15	11.0–18.1	[15,49,50]
Hundred-seed weight	42	5.68–35.9	[15,31,34,63,89,92]
Haulm weight	11	2.9–33.36	[15,48,92]
Pod number	24	3.91–14.2	[15,32,93]
Total biomass	15	4.34–22.39	[15,48,50]
Growth habit	48	4.55–27.14	[15,64,96]
**Drought-Tolerance-Related Traits**			
Traits Studied	QTL Identified	Phenotypic Variance Explained	References
Shoot dry weight	16	4.2–22.09	[49,50]
Transpiration efficiency	27	4.47–18.12	[48,49,50]
Leaf area	26	5.0–16.2	[48,50]
Transpiration rate	13	4.3–17.3	[50]
Transpiration	16	4.36–18.17	[48,49]
SPAD chlorophyll meter readings (SCMR)	29	4.00–19.53	[48]
Stress Tolerance Index	13	10.4–20.1	[15]
**Pest- and Disease-Related Traits**			
Traits Studied	QTL Identified	Phenotypic Variance Explained	References
Nematode	7	1.3–22.18	[17]
Leaf spot	80	1.7–50.9	[51,53,54,55,62,63,97]
Bacterial wilt	3	0.12–0.22	[58]
Rust	34	7.24–48.7	[52,55,62,63]
Smut	10	7.24–11.4	[56]

## Data Availability

The data used in this study are available through the Appendix A. No new data were created.

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
