# Peer review of "An Overview of Mapping Quantitative Trait Loci in Peanut (Arachis hypogaea L.)"

_genes, 2023, doi:10.3390/genes14061176_

Round 1

Reviewer 1 Report

In this study, the authors reviewed the recently progress on peanut Quantitative Trait Loci (QTL) analysis with a particular emphasis on mapping populations and the trait related to kernel quality. The authors found that several types of populations including interspecific populations have been used for QTL mapping and elite alleles identifying for agronomic traits in peanut. It was found that among the 1,261 QTLs reported in this review and the relevant published articles on peanut QTL mapping, 413 QTLs were related to the kernel quality, indicating the importance of quality in peanut breeding. This study would provide valuable information for exploiting QTLs and molecular marker-assisted breeding in peanut. I suggest acceptance of the manuscript in present form.

Author Response

The first reviewer made very valuable comments without revisions. All  authors thank the reviewer for agreeing to examine the paper and frutfully appreciate and approuved his valuable comments.

Reviewer 2 Report

This manuscript was well written reviewing QTLs identified in peanut. The authors collected a lot of useful information on DNA markers used, mapping populations generated, and QTLs colocalized on the same chromosomal regions. This review will be useful for the peanut community. To enhance the quality of this manuscript, I have the following comments and suggestions:

Major revision:

Figure 3 presented very useful information for thirteen seed quality traits. The readers can clearly see the colocalizations of some QTLs from different reports. These colocalizations can potentially help researchers to identify the candidate genes (QTLs) for the seed quality traits and then develop functional SNP markers.

From Table 2 now, we only can see how many QTLs identified for the agronomic traits but without chromosomal locations.  I suggest authors should generate a similar figure like Figure 3 and summarize the results.

Minor revisions:

Page 2 lines 51-52, what is the difference between palm kernel and palm oil. It was not clear to me. From literature, peanut is recognized as the fifth oilseed crop in the world, but the authors stated as the sixth oilseed crop. Please double check the literature.

Line 950, reference 118, please use abbreviations for the journal name.

Author Response

  • Point 1: Suggestion of the Reviewer to generate a similar Figure like Figure 3 for QTL related to agronomic traits

Response 1 of Authors: We thank the referee’s suggestion for a Figure of chromosomal region of QTL related to agronomic traits. As far as QTLs for agronomic traits are concerned, the number of the studies that reported QTLs and the lack of common markers between studies that are needed to bridge several individual maps made it difficult to draw a consensus QTL maps with relevant information.  This is the main reason why we described the results as table rather than as map for a figure of chromosomal region of QTL related to agronomic traits. Please note that we have tried few months ago to draw a map of agronomic QTL, as well as, were draw QTL of quality traits. The result was less relevant and did not provide more information compared to the specific maps already published, as mentioned in Table 2. The result was not relevant since there were little common QTL and less common markers among studies that help to bridge and draw a map showing those chromosomal regions related to agronomic traits. In addition, except for few agronomic traits like Hundred pod/seed weight, there are little relevant common QTL among studies related to agronomic traits. Furthermore, with the aim to be objective and specific, we provided a Figure 3 of QTL related to quality traits since the review focused on quality traits with a particular emphasis on traits related to kernel quality, indicated at lines 110-111, in accordance to the topic of the manuscript . However, we have specified in the text manuscript some all known cases of co-localization between relevant agronomic and quality trait QTLs that co-locate with each other and with those of quality traits. This is summarized in the text at the point 4.5 Clustering of QTLs of quality and agronomic traits” from line 475 to 507.

  • Point 2 : Revision 1, Fixed

Reviewer: Page 2 lines 51-52, what is the difference between palm kernel and palm oil. It was not clear to me. From literature, peanut is recognized as the fifth oilseed crop in the world, but the authors stated as the sixth oilseed crop. Please double check the literature.

Response 2 of Authors: The referee is right about the confusion regarding palm kernel and palm oil. We erroneously considered the oil of the palm seed (palm kernel) and the oil of the palm fruit (palm oil) separately, which has now been rectified. In relation to world ranking of peanut among oilseed crops, we have revised it from “sixth” to “fifth” according to the referee’s comment.

  • Point 3: Revision 2, Fixed

Reviewer: Line 950, reference 118, please use abbreviations for the journal name.

Response 3 of Authors: We checked and used the abbreviation of the journal name according to the referee comment and the Gene journal instructions

Finally, we fixed the two revisions. All authors thank the reviewer for agreeing to examine the paper and fruitfully appreciate and approve his valuable comments.